# Hardware Implementation for Triaxial Contact-Force Estimation from Stress Tactile Sensor Arrays: An Efficient Design Approach

**DOI:** 10.3390/s24237829

**Published:** 2024-12-07

**Authors:** María-Luisa Pinto-Salamanca, Wilson-Javier Pérez-Holguín, José A. Hidalgo-López

**Affiliations:** 1Programa de Doctorado en Ingeniería-Énfasis en Ingeniería Electrónica, Grupo GIRA, Universidad Pedagógica y Tecnológica de Colombia UPTC, Sogamoso 152211, Colombia; wilson.perez@uptc.edu.co; 2Programa de Doctorado en Ingeniería Mecatrónica, Departamento de Electrónica, Universidad de Málaga, 29071 Malaga, Spain; jahidalgo@uma.es

**Keywords:** tactile sensing, triaxial contact forces estimation, hardware implementation, sparse matrix-vector multiplication, FPGA

## Abstract

This paper presents a contribution to the state of the art in the design of tactile sensing algorithms that take advantage of the characteristics of generalized sparse matrix-vector multiplication to reduce the area, power consumption, and data storage required for real-time hardware implementation. This work also addresses the challenge of implementing the hardware to execute multiaxial contact-force estimation algorithms from a normal stress tactile sensor array on a field-programmable gate-array development platform, employing a high-level description approach. This paper describes the hardware implementation of the proposed sparse algorithm and that of an algorithm previously reported in the literature, comparing the results of both hardware implementations with the software results already validated. The calculation of force vectors on the proposed hardware required an average time of 58.68 ms, with an estimation error of 12.6% for normal forces and 7.7% for tangential forces on a 10 × 10 taxel tactile sensor array. Some advantages of the developed hardware are that it does not require additional memory elements, achieves a 4× reduction in processing elements compared to a non-sparse implementation, and meets the requirements of being generalizable, scalable, and efficient, allowing an expansion of the applications of normal stress sensors in low-power tactile systems.

## 1. Introduction

In artificial tactile sensing systems, the measurement of contact forces plays a crucial role in the description of the contact phenomenon [1,2] seeking to resemble human dexterity for object manipulation [3]. Force estimation provides direct information about the contact area and facilitates the reconstruction of other tactile properties such as roughness [4,5], and hardness [6]. Moreover, triaxial forces are critical for slip and grip sensing in control loops for robotic manipulation [3,4,7,8,9,10,11,12], wearable robots [13], human-computer interaction [2], prosthetic hands and wearable devices [14,15,16].

The scientific community has already provided solutions for reconstructing normal, shear, and multiaxial contact forces. Some authors [12,17,18,19] have estimated forces using integrated biomimetic-design tactile sensors for robotic grippers using vision-based transducers. Alternatively, the piezoelectric effect [20,21], piezoresistive-based tactile sensor arrays [2,14,15,22], capacitive sensors [13,16,23,24,25,26,27], magnetic transducers [28,29], optical fibers [30,31], and Hall effect transducers [32] can be used. However, contact-force sensing remains a major research challenge due to issues such as the complexity and multiplicity of mathematical force estimation models, the different transduction technologies employed, the commercial availability of sensors, the area size and contact medium, and the demand for real-time processing [2,33].

Despite recent advances in multiaxial contact-force reconstruction, the community continues to work towards achieving tactile sensing systems that meet the criteria of generalization, scalability, and efficiency described in [34]. A challenge to generalization lies in the fact that force estimation methods depend on the information provided by the tactile sensor and the contact medium used. Scalability is limited by the transduction technology and sensor size used with contact areas typically small (less than 164 mm × 164 mm) [25,35]. In terms of efficiency, force reconstruction time must be predictable and done in real time [2,13,15,23,25]. All of this involves systems with high processing requirements that need very short response times. The approach to these issues is based on the development of specialized software that runs on high-performance PC stations, which limits their application in cases where high portability is required, such as in electronic skin applied in biomedical or robotic systems.

Regarding sensor types used in tactile sensing applications, pressure-sensor arrays offer advantages such as high resolution, high sensitivity, low noise, simple electronics, physical flexibility, large surface area, wide commercial availability and price, dynamic response, low thickness, large-scale coverage, and high bandwidth [36,37,38,39,40]. In spite of their advantages, the use of pressure-sensor arrays in multiaxial force reconstruction has been quite limited, as they only provide information on normal stresses and require complex mathematical models to process the data they generate. Their applications have been limited to obtaining normal forces [37], and through subsequent processing stages, other authors have estimated shear [39] and triaxial forces [41], but none of those methods have been validated in hardware using efficient performance metrics. In [42], an optimized option for the hardware implementation of triaxial force estimation in real-time tactile decoding tasks is provided. However, since such implementation has only been validated in software, it is necessary to investigate the implications of the properties of its mathematical model on hardware design, memory consumption, and data-processing efficiency.

An alternative to address the high complexity, real-time execution, and large data volume requirements of tactile decoding systems is to leverage the benefits of highly parallel embedded systems such as field-programmable gate arrays (FPGAs), graphics processing units (GPUs), and others. These systems provide high concurrency, lower power consumption, and high computational acceleration, which, if properly exploited through a highly optimized design, can greatly exceed the performance of software-based (PC) solutions [43]. Embedded systems have been used to implement model- and data-driven solutions in tactile sensing applications like texture estimation [44], texture classification [45], tactile data decoding [46,47], slip detection [48], and force estimation [41]. However, hardware implementations in this context are scarce because the hardware design process remains slow and complex [43], is highly dependent on the sensor characteristics, transduction technology, and contact type, and the mathematical model is not always easy to modify as it involves significant changes in the hardware deployment. One way to overcome these problems is to use high-level design methodologies, which have been validated in other fields but rarely used to develop hardware for tactile sensing systems [49].

This work contributes to the theoretical development of algorithms for multiaxial contact-force reconstruction from normal stress tactile sensor arrays by proposing using sparse matrices to reduce the computational requirements for its implementation in hardware in terms of area, power consumption, and data storage. It also contributes to the practical implementation of tactile sensing algorithms in hardware by comparing the results of [42] with those obtained for a new model presented herein. Although it has only been validated for two single static contacts on flat surfaces, we believe this work will contribute to expanding the use of stress sensor arrays for multiaxial force estimation in large-scale wearable electronic skin systems.

Among the most relevant aspects of the hardware implementation carried out here are that it is efficient (in terms of computation times and area occupation), generalizable (independent of the transduction sensor technology and the contact phenomenon), scalable (independent of the size of the contact sensing area) and described at a high level (using high-level tools such as Matlab^®^ MathWorks™, AMD Vitis™ HLS, and Vivado™ Design Suite) which allows a reduction of the design effort and quick validation of the functionality of the system.

The rest of the paper is organized as follows: Section 2 focuses on recalling the main features of the reconstruction forces algorithm addressed, Section 3 describes the strategy of integrating sparse matrices into the forces algorithm, Section 4 presents the hardware design flow and the efficiency criteria considered, Section 5 discusses the results obtained, and Section 6 concludes the paper.

## 2. Triaxial Forces Reconstruction Algorithm from Normal Stress Tactile Sensor Arrays

In the field of tactile property decoding, the TFRA algorithm [42] stands out for allowing the reconstruction of triaxial forces in large contact areas from tactile sensor arrays. Such an algorithm inputs a vector with scalar values of normal stresses and produces as output three vectors with the triaxial force magnitudes (one for each axis), which correspond to a solution to the problem of estimating multiaxial forces from a discrete normal stress distribution as shown in Figure 1. In addition, it is characterized by being generalizable, efficient, and scalable, which makes its implementation feasible for real-time tactile decoding systems.

The TFRA algorithm is based on the inverse problem of the classical Boussinesq equation, which determines the stress fields in a homogeneous and linearly elastic half-space under a concentrated load applied at a specific point on its upper surface [50]. The relationship between stress fields and load forces, established by the Boussinesq equation, is very significant for contact modeling on flat surfaces using tactile sensing arrays [41] since if the contact load is concentrated into an array of *f* point forces, then a *b* stress discrete distribution can be calculated through a linear vector equation as:(1)b=Cf
where *C* is a rectangular matrix defined by the distances between the coordinates of the *m*-stress points and those of the *n*-force vectors, $b={\left[b_{x},b_{y},b_{z}\right]}^{T}$ is a vector of size [3m×1] with $b_{x}$ and $b_{y}$ represents the tangential stress, while $b_{z}$ is the normal stress, and $f={\left[f_{x},f_{y},f_{z}\right]}^{T}$ is a vector of size [3n×1] of $f_{x}$, $f_{y}$, and $f_{z}$ components. Under the premise of Equation (Equation 1), if m=n, *C* is a square full-rank matrix, and the direct application of this equation generates *m*-values of the spatial stress distribution from *m*-triaxial force vectors; see Figure 2.

For the same case, the inverse application of Equation (Equation 1), expressed as:(2)$$f=C^{-1}b$$
This allows the discovery of *m*-force vectors from the stress values at *m*-points, and models the contact forces on a tactile sensor array of *m*-discrete sensing units (taxels), with m=u×v, where *u* is the number of rows and *v* the number of columns of the array; see Figure 3. For this problem, the application of Equation (Equation 1) allows us to find the normal stress distribution at the *m*-points located at a depth *h* of the surface (sensor thickness), defined as:(3)$$b_{z}=\left[\begin{array}{ccc} C_{31} & C_{32} & C_{33} \end{array}\right]\left[\begin{array}{c} f_{x}\\ f_{y}\\ f_{z} \end{array}\right]$$
with $C_{31}$, $C_{32}$, and $C_{33}$ constant submatrices of size [m×m] defined from the sensor geometry characteristics, as:(4)$${\left[\begin{array}{ccc} C_{31} & C_{32} & C_{33} \end{array}\right]}^{T}=\left[\begin{array}{c} C_{0}\cdot \hat{r}{\left(ji\right)}_{x}\\ C_{0}\cdot \hat{r}{\left(ji\right)}_{y}\\ C_{0}\cdot h \end{array}\right]$$
where $C_{0}=\left(3h^{2}\right)/\left(2\pi {\left(\hat{r}{\left(ji\right)}^{2}+h^{2}\right)}^{5/2}\right)$, $\hat{r}\left(ji\right)$ is the projection on the xy-plane of the r(ji) distance vectors between the coordinates $\left(\hat{r}{\left(ji\right)}_{x},\hat{r}{\left(ji\right)}_{y}\right)$ of the *i*-th taxel (i=1,…,m) and the *j*-th force vector (j=1,…,m) [51], see Figure 3b.

For the case of a tactile sensor that only provides discrete normal stress data $b_{z}$ for the *m*-taxels, the inverse problem becomes ill-posed because the number of unknowns in Equation (Equation 2) is three times larger than the sensor outputs, requiring an additional mathematical description. In such a case, the Moore–Penrose pseudo-inverse matrix provides a solution of the form $f=A^{-1}\cdot b_{z}$ [52]:(5)$$f={\left[\begin{array}{ccc} f_{x} & f_{y} & f_{z} \end{array}\right]}^{T}=A_{†}b_{z}+\left(I-A_{†}A\right)w$$
where *A* is a rectangular matrix of size [m×3m], defined as:(6)$$A=\left[\begin{array}{ccc} C_{31} & C_{32} & C_{33} \end{array}\right]$$

$A_{†}$ is the pseudo-inverse matrix of *A* of size [3m×m], such that:(7)$$A_{†}=A^{T}{\left(AA^{T}\right)}^{-1}={\left[A_{†1}A_{†2}A_{†}3\right]}^{T}$$
in which $A_{†1}$, $A_{†2}$, and $A_{†3}$ are constant submatrices of size [m×m], and *w* is an optimal vector of size [m×1], for which
(8)$$w=\left[\begin{array}{c} \mu_{x}\left|w_{z}\right|\\ \mu_{y}\left|w_{z}\right|\\ w_{z} \end{array}\right]$$
where $\mu_{x}$ and $\mu_{y}$ are two continuous scalars in the range [−1,1] defined into an optimization process, and $w_{z}$ a vector made up of the negative values of the tactile sensor, as follows:(9)$$w_{z}=\left\{\begin{array}{cc} C_{33}^{-1}\cdot b_{z} & ;\left(C_{33}^{-1}\cdot b_{z}\right)\left(i\right)<0\\ 0 & ;\mathrm{other}\;\mathrm{case} \end{array}\right.$$

By replacing Equations (Equation 6)–(Equation 9) in Equation (Equation 5), the TFRA allows the discovery of an optimal solution for this problem by calculating the components of a *f* triaxial forces as follows:(10)$$\begin{array}{c} \left[\begin{array}{c} f_{x}\\ f_{y}\\ f_{z} \end{array}\right]=\left[\begin{array}{c} g_{10}+\mu_{x}g_{11}+\mu_{y}g_{12}\\ g_{20}+\mu_{x}g_{21}+\mu_{y}g_{22}\\ g_{30}+\mu_{x}g_{31}+\mu_{y}g_{32} \end{array}\right] \end{array}$$
where $g_{pq}$ vectors of size [m×1] are computed as shown in Equations (Equation 11)–(Equation 13) for p=[1:3], q=[0:2], such that:(11)$$\begin{array}{c} {\left[\begin{array}{ccc} g_{10} & g_{11} & g_{12} \end{array}\right]}^{T}=\left[\begin{array}{c} A_{†1}b_{z}+D_{10}w_{z}\\ |w_{z}|+D_{11}|w_{z}|\\ D_{12}|w_{z}| \end{array}\right] \end{array}$$
(12)$$\begin{array}{c} {\left[\begin{array}{ccc} g_{20} & g_{21} & g_{22} \end{array}\right]}^{T}=\left[\begin{array}{c} A_{†2}b_{z}+D_{20}w_{z}\\ D_{21}|w_{z}|\\ |w_{z}|+D_{22}|w_{z}| \end{array}\right] \end{array}$$
(13)$$\begin{array}{c} {\left[\begin{array}{ccc} g_{30} & g_{31} & g_{32} \end{array}\right]}^{T}=\left[\begin{array}{c} A_{†2}b_{z}+w_{z}+D_{30}w_{z}\\ D_{31}|w_{z}|\\ D_{32}|w_{z}| \end{array}\right] \end{array}$$

$D_{pq}$ are nine matrices of size [m×m] calculated by an offline precomputation carried out by matrix multiplications between the submatrices of $A_{†}$ and *C*, as follows:(14)$$\begin{array}{c} \left[\begin{array}{ccc} D_{10} & D_{11} & D_{12}\\ D_{20} & D_{21} & D_{22}\\ D_{30} & D_{31} & D_{32} \end{array}\right]=\left[\begin{array}{ccc} -A_{†1}C_{33} & -A_{†1}C_{31} & -A_{†2}C_{32}\\ -A_{†2}C_{33} & -A_{†2}C_{31} & -A_{†2}C_{32}\\ -A_{†3}C_{33} & -A_{†3}C_{31} & -A_{†3}C_{32} \end{array}\right] \end{array}$$

To implement Equation (Equation 10), TFRA requires the execution of 16 matrix-vector multiplications, denoted as $M_{0}-M_{15}$, and the iterative calculation of the optima of $\mu_{x}$ and $\mu_{y}$. The first 13 operations ($M_{0}-M_{12}$) correlate the sensor geometry and the normal stress distribution measured by the tactile sensor array so that: (15)$$M_{0}=C_{33}^{-1}b_{z}$$
(16)$$\left[\begin{array}{ccc} M_{1} & M_{2} & M_{3}\\ M_{4} & M_{5} & M_{6}\\ M_{7} & M_{8} & M_{9}\\ M_{10} & M_{11} & M_{12} \end{array}\right]=\left[\begin{array}{ccc} D_{10}w_{z} & D_{11}|w_{z}| & D_{12}|w_{z}|\\ D_{20}w_{z} & D_{21}|w_{z}| & D_{22}|w_{z}|\\ D_{30}w_{z} & D_{31}|w_{z}| & D_{32}|w_{z}|\\ A_{†1}b_{z} & A_{†2}b_{z} & A_{†3}b_{z} \end{array}\right]$$

The last three operations ($M_{13}-M_{15}$) produce $b_{z}^{z}$ and $b_{z}^{xy}$ vectors to evaluate the fulfillment of the optimal conditions:(17)$$\left[\begin{array}{ccc} M_{13} & M_{14} & M_{15} \end{array}\right]=\left[\begin{array}{ccc} C_{33}f_{z}^{k} & C_{31}f_{x}^{k} & C_{32}f_{y}^{k} \end{array}\right]$$
where $f_{x}^{k}$, $f_{y}^{k}$, and $f_{z}^{k}$ are force components estimated for the k-th iteration of the algorithm, while the vectors $b_{z}^{z}$ of size [m×1], and $b_{z}^{xy}$ of size [2m×1] are defined as:(18)$$b_{z}^{z}=C_{33}f_{z}^{k}$$
(19)$$b_{z}^{xy}=\left[\begin{array}{cc} C_{31} & C_{32} \end{array}\right]\left[\begin{array}{c} f_{x}^{k}\\ f_{y}^{k} \end{array}\right]$$

As mentioned in [42], the TFRA optimization process allows us to ensure that: (i) normal forces ($f_{z}^{k}$) will be correctly reconstructed if they generate only a compression effect at the sensor base when they are considered independently, and (ii) tangential forces ($f_{x}^{k}$, $f_{y}^{k}$) will be correctly reconstructed if they generate a traction distribution similar to that measured by the sensor. Formally, each condition is evaluated as follows:

Condition 1:(20)$$B^{z}\equiv \sum_{i=1}^{m}b_{z}^{z}\left(i\right)=0;b_{z}^{z}\left(i\right)>0$$

Condition 2:(21)$$\begin{array}{c} B^{xy}\equiv \left|\sum_{i=1}^{m}b_{z}^{xy}\left(i\right)-\sum_{i=1}^{m}b_{z}\left(i\right)\right|=0;\\ b_{z}^{xy}\left(i\right)>0,b_{z}\left(i\right)>0 \end{array}$$

To evaluate the optima $\mu_{x}$ or $\mu_{y}$ in Equation (Equation 10), TFRA selects only one of these, $\mu_{x}$ or $\mu_{y}$, as an independent variable and evaluates the other (the unselected one) from the angle between the tangential forces (ϕ) and the $G_{pq}$ resultants of the vectors $g_{pq}$. Therefore, the dependent variable selected can be computed as:(22)$$\mu_{x}=\frac{G_{20}-G_{10}\tan\phi +\mu_{y}\left(G_{22}-G_{12}\tan\phi \right)}{G_{11}\tan\phi -G_{21}}$$

or
(23)$$\mu_{y}=\frac{G_{10}\tan\phi -G_{20}+\mu_{x}\left(G_{11}\tan\phi -G_{21}\right)}{G_{22}-G_{12}\tan\phi }$$
where $G_{pq}=\sum_{i=1}^{m}g_{pq}\left(i\right)$, and ϕ is calculated by identifying the compression centroids, with coordinates $\left(c_{x},c_{y}\right)$, and the tension centroids, with coordinates $\left(t_{x},t_{y}\right)$, of the stress distribution, as:(24)$$\phi =\left\{\begin{array}{cc} \tan^{-1}\left(\frac{c_{y}-t_{y}}{c_{x}-t_{x}}\right) & ;t_{x}<c_{x}\\ \tan^{-1}\left(\frac{c_{y}-t_{y}}{c_{x}-t_{x}}\right)+180^{\circ } & ;t_{x}\geq c_{x} \end{array}\right.$$

Following the aforementioned specifications, TFRA requires six stages (see Figure 4) to evaluate Equation (Equation 10) for:i.  [For one time only (offline stage)] Calculate and save in memory the set of matrices ($A_{†1}$, $A_{†2}$, $A_{†3}$, $C_{33}^{-1}$, $D_{pq}$, $C_{31}$, $C_{32}$, and $C_{33}$), the vectors describing the taxel coordinates $\left(p_{x},p_{y}\right)$, and the sensor resolution $\left(s_{x},s_{y}\right)$,ii. Read and store in memory the data for sensor $b_{z}$ as an m×1 size vector,iii.Calculate the contact centroids and angle ϕ of tangential forces,iv. Calculate the $w_{z}$ vector,v.  Calculate the $g_{pq}$ coefficient vectors,vi. Find the optima $\mu_{x}$ and $\mu_{y}$, and compute the forces triaxial reconstruction.

Equations (Equation 15)–(Equation 17) are computed in stages iv–vi, requiring 16 matrix-vector multiplications with a computational complexity of $O(m^{2})$ for *m*-taxels in a tactile sensor array. When in a software implementation, these algebraic operations can be easily performed. In the case of a dedicated hardware implementation, design decisions must include processing requirements, memory size, and multiple memory accesses in advance. Thus, the hardware design process to implement the TFRA involves 16m accumulates, $16m^{2}$ data store operations, $16m^{2}$ memory accesses to a constant array, and 16m memory accesses to vector data.

## 3. SpTFRA Algorithm for Contact Forces Reconstruction

This section introduces a new algorithm, named SpTFRA (Sparse Triaxial Forces Reconstruction Algorithm), that allows the improvement of the performance of the TFRA algorithm by reducing the hardware resource consumption and the number of operations required for its calculation through the use of sparse matrices. The software implementation of the TFRA algorithm showed that in constant matrices $A_{†1}$, $A_{†2}$, $A_{†3}$, $C_{33}^{-1}$, and $D_{pq}$(computed by Equations (Equation 5), (Equation 7) and (Equation 14)), their components with the highest values are concentrated in coordinates close to their diagonals (see Figure 5) and exhibit a data distribution similar to sparse matrices (mostly zeros), a behavior demonstrated in different contact cases. This can be attributed to the fact that the components of these matrices are defined by the relative distances between the coordinates of the *i*-th taxel and the *j*-th estimated force, meaning that the force vectors are mainly influenced by nearby taxels in the xy-plane.

Matrix–vector multiplication is a basic operation required in many physical systems, so its optimization is of interest in several scientific domains. The generalized sparse-matrix dense-vector multiplication (SpMv) is defined as y=Bx, where *B* is a sparse matrix and *x* is a dense vector. This multiplication can be efficiently compressed using data compression and encoding strategies that store only the non-zero (Nnz) values of the matrix and its coordinates.

One of the challenges to the hardware implementation of matrix operations relates to the on-chip and off-chip memory access [53] and the design of processing blocks that exactly suit the distribution of zeros in sparse matrices. Some algorithms to optimize SpMv on hardware have already been studied for large-scale matrix dimension problems in high-performance computing for physical or biological model simulations [54,55], data analytics [56], large-scale graphics processing [57], and artificial intelligence [58,59]. These algorithms provide solutions on GPU, FPGA, or heterogeneous architectures to process matrices as large as 28,338 × 28,338, and response times less than 5 ms [60].

In SpTFRA, it is assumed that all values of sparse-like matrices close to zero are effectively zero. This may slightly increase the error, but it reduces the resources required for data storage and the number of operations to be performed. In this way, the implementation of the SpTFRA algorithm requires the use of SpMv operations for the calculation of the matrix–vector multiplications $M_{0}-M_{12}$ defined by Equations (Equation 15) and (Equation 16) in which the set of matrices ($A_{†1}$, $A_{†2}$, $A_{†3}$, $C_{33}^{-1}$, and $D_{pq}$) is offline replaced with a set of approximate sparse matrices. Multiplications $M_{13}$, $M_{14}$, and $M_{15}$ were already included in the Op block of the original TFRA as sparse operations, so no modifications are needed for the SpTFRA implementation.

Although SpTFRA uses the same six stages described in Section 2, there are three main differences between both the TFRA and SpTFRA algorithms: (i) calculating and storing the sparse matrices in the memory block, (ii) executing $M_{0}$ as SpMv in the We block, and (iii) executing $M_{1}-M_{12}$ as SpMv operations in the Co block. All other functional blocks are common for both algorithms (see Figure 6).

To implement SpTFRA, the proposed approach requires applying three filters that make zero some components of a matrix if its magnitude is less than a previously determined threshold. This allows the obtaining of an approximate sparse matrix $B_{sp}$(i.e., $A_{†1sp}$, $A_{†2sp}$, $A_{†3sp}$, $C_{33sp}^{-1}$, and $D_{pq-sp}$) from a dense matrix *B*($A_{†1}$, $A_{†2}$, $A_{†3}$, $C_{33}^{-1}$, and $D_{pq}$). The threshold value is set under three specific criteria, named Sparse Filters, which are depicted in Figure 7, and defined as follows:SpTFRA−F1: This filter admits only the non-zero values B(i,j) closest to the diagonal in a dense matrix *B*, by evaluating the function:
(25)$$B_{sp1}\left(i,j\right)=\left\{\begin{array}{cc} B(i,j) & if|i-j|\leq L\\ 0 & other-case \end{array}\right\}$$
where *L* is a value that establishes the expanded diagonal matrix made up of the components of *B* that satisfy |i−j|≤L, with *L* changing at each experiment by 10 in a range from 10 to 90 (according to the estimation error calculated by the SpTFRA). Figure 8 shows an example of applying this filter on the $C_{33}^{-1}$ matrix, where non-zero values are in blue.SpTFRA−F2: This filter accepts the components greater or equal to a percentage *p* of the maximum value for each row p∗max(B(i,:)), by evaluating the function,
(26)$$B_{sp2}\left(i,j\right)=\left\{\begin{array}{cc} B(i,j) & if|B_{ij}|\geq \left(1/100p\right)*max\left(B\left(i,:\right)\right)\\ 0 & other-case \end{array}\right\}$$
where *p* changes by 10 in a range from 10 to 90 at each experiment. Figure 9 shows the results of applying this filter to the $C_{33}^{-1}$ matrix.SpTFRA−F3: This filter selects between Filters 1 and 2, depending on which of these generates the sparse matrix Bsp with the minimum value of Nnz and the minimum error in the estimation of the contact forces, according to this function:
(27)$$B_{sp}\left(i,j\right)=\left\{\begin{array}{cc} B_{sp1}\left(i,j\right) & e_{\mu }\left(B_{sp1}\right)Nnz\left(B_{sp1}\right)\leq e_{\mu }\left(B_{sp2}\right)Nnz\left(B_{sp2}\right)\\ B_{sp2}\left(i,j\right) & other-case \end{array}\right\}$$
where $e_{\mu }$ is the error estimate of the friction coefficient in a force reconstruction and comes defined as follows:
(28)$$\begin{array}{c} e_{\mu }=100\frac{\mu_{ref}-\mu }{1.0}\% \end{array}$$
with μ calculated as:
(29)$$\mu =\frac{\sqrt{F_{x}^{2}+F_{y}^{2}}}{|F_{z}|}$$
in which $F_{x}=\sum_{i=1}^{m}f_{x}\left(i\right)$ and $F_{y}=\sum_{i=1}^{m}f_{y}\left(i\right)$ are the estimated resultant tangential forces, and $F_{z}=\sum_{i=1}^{m}f_{z}\left(i\right)$ is the reconstructed resultant normal force.

After filtering the matrices of the TFRA, the modified compressed sparse row (MCSR) format [60] is used to store the compressed matrix $B_{sp}$ in memory and calculate the SpMv on three vectors, which include the Nnz values of the matrix $B_{sp}$, the column corresponding to each Nnz in $B_{sp}$, and the number of multiplication accumulations (MAC) for each row. Due to the MCSR representation, the matrix–vector multiplication algorithm does not require adding or multiplying by zero, which means saving arithmetic operations proportional to the Nnz of each matrix and reducing memory consumption when storing SpTFRA matrices.

## 4. Hardware Design for the TFRA and SpTFRA Implementations

The design of the hardware architecture for the TFRA and SpTFRA are based on the four subsystems (CeAn, We, Co, and Op) shown in Figure 4 and correspond to the six stages of the force reconstruction algorithm described in Section 2. The development of the architecture was carried out by evaluating (in software) the functionality of the proposed algorithms. Then, it was implemented in hardware (on a development platform for FPGA) following the classic integrated circuit design flow presented in [61].

To evaluate the functionality and efficiency of the TFRA and SpTFRA hardware implementations, we established two criteria:

*Algorithm functionality*: TFRA and SpTFRA should be generalizable (i.e., they should depend only on the contact event but not on the transduction technology), scalable (i.e., operate over different contact area sizes), and efficient (i.e., have a predictable runtime) as established in [34]. For the experimental verification of this criterion, we calculate the forces estimation error and the response time, evaluating by simulation, two cases of simple contacts (Hertzian and non-Hertzian) using two pressure-sensor arrays of different resolutions:
Sensor 1. It is a 10×10 taxel array with a $\left(s_{x},s_{y}\right)=\left(4\;\mathrm{mm},2\;\mathrm{mm}\right)$ resolution arranged as a rectangular prism of dimensions 40mm×20mm×3mm, on which a single static Hertzian contact was modeled, equivalent to a distribution of elliptical normal stresses measured at the base of the tactile sensor and centered on the coordinates $\left(x_{0},y_{0}\right)=\left(18\;\mathrm{mm},9\;\mathrm{mm}\right)$.Sensor 2. It is a 10×10 taxel array with a $\left(s_{x},s_{y}\right)=\left(4\;\mathrm{mm},4\;\mathrm{mm}\right)$ resolution arranged as a rectangular prism of dimensions 40mm×40mm×3mm, on which a single static non-Hertzian contact was modeled measured at the base of the tactile sensor and centered in the coordinates $\left(x_{0},y_{0}\right)=\left(18\;\mathrm{mm},18\;\mathrm{mm}\right)$.

The TFRA and SpTFRA software implementations were carried out in Matlab^®^ MathWorks™ 2021. To evaluate the estimation error and response time, we compared the Matlab results with those obtained for a finite element analysis (FEA) model implemented in COMSOL Multiphysics^®^ 6.0.

*Hardware Efficiency*: A hardware efficient implementation of TFRA and SpTFRA should perform triaxial force reconstruction with the lowest error estimation, power consumption, hardware resource usage and memory requirements, the highest throughput, and the shortest response time. That is why, in the design verification process, we consider the corresponding metrics for each aspect and select the best alternative.

The entire hardware design process was carried out on AMD Vitis™ HLS 2022 and Vivado™ Design Suite 2022 running on a laptop based on an Intel Dual-Core i7-4600U CPU at 2.1 GHz and 8 GB of RAM. The SpTFRA hardware implementation was evaluated on a Zynq UltraScale+ MPSoC ZCU102 FPGA. Both cases used a 32-bit floating-point format and a system clock frequency of 125 MHz.

### High-Level Desing Approach of the TFRA and SpTFRA

High-level synthesis (HLS) is a modern and efficient automated design process for digital systems that focuses on mapping behavioral/algorithmic specifications (in C/C++ or SystemC) on register-transfer level (RTL) structures. This technique speeds up the verification procedures at early stage designs, supporting hardware designers to improve functional features while tuning up optimization targets [62], considerably reducing the design effort and supporting the exploration of test and reliability analyses [49,63].

This work exploits the advantages of an HLS tool to implement the TFRA and SpTFRA algorithms in hardware. Before that, a software-level description ($SW_{SF(L|p)}$) was used to better understand the behavior of such algorithms and predict their characteristics at the hardware level ($HW_{SF(L|p)}$) when using the sensors *S* (S:1−2), the filters *F* (F:1−3) and performing a sweep on their variables *L* (L:10:10:90) or *p* (p:10:10:90) for each filter.

The following nomenclature was used for the TFRA software ($SW_{1}$, $SW_{2}$), and hardware ($HW_{1}$, $HW_{2}$) implementations, depending on the type of sensor (S:1−2). Similarly, for the SpTFRA, the software and hardware implementations were named $SW_{SF(L|p)}$ and $HW_{SF(L|p)}$, respectively, depending on the type of sensor (S:1−2) and filter used (F:1−2). For example, $SW_{1F1(L10:L90)}$ represents Sensor 1 with Filter 1 and a sweep of L from 10 to 90. Meanwhile, $SW_{1F2(p10:p90)}$ represents Sensor 1 with Filter 2, with a sweep of p from 10 to 90. Finally, the SpTFRA implementations for the case of Filter 3 were named $SW_{1F3}$, $SW_{2F3}$ and $HW_{1F3}$, $HW_{2F3}$, according to the type of sensor used.

The testbench was also implemented in HLS, allowing for reports on hardware synthesis, latency, resources, and power consumption. From test bench results, it was possible to reduce from 42 hardware implementations to only four (two for each sensor).

## 5. Results

### 5.1. Sparse Filters Response

Figure 10 shows the friction coefficient μ obtained in the TFRA and SpTFRA software implementations for a reference value $\mu_{ref}=0.5$ and variations in the angle of the orientation of the tangential forces in the range of $\phi =[0–360^{\circ }]$ for the Hertzian (Sensor 1) and non-Hertzian (Sensor 2) single contacts modeled. μ was computed at each ϕ value from the triaxial forces reconstructed by applying Equation (Equation 29).

Figure 11 summarizes the behavior of all the filters proposed in Section 3 for each sparse matrix in both algorithms. The values presented were calculated from Equations (Equation 27)–(Equation 29), in which the number of Nnz is correlated with the estimation error of μ obtained for each filter. In this figure, the best cases are given for the cells with the lowest values, as this corresponds to the minimum error and Nnz. This leads to sparser matrices that require both less memory and fewer processing elements. Similarly, the white-on-orange text cells in Figure 11 represent the best cases for the SpTFRA−F3 filter implementation.

Table 1 describes the filters selected at the hardware design stage, their Nnz values, and the estimation of the average friction coefficient due to the variation of ϕ. The best reconstruction cases were obtained for L=40 and L=50 at SpTFRA−F1 and p=30 and p=50 at SpTFRA−F2 for both sensors.

### 5.2. Hardware Resource Consumption

From the information presented in Table 1, the use of digital signal processors (DSP), block random access memory (BRAM), look-up tables (LUT), and Flip-Flops (FF) elements required by each hardware implementation was compared. The results of this comparison are shown in Figure 12. As can be seen, the SpTFRA requires considerably fewer DSP blocks than the original TFRA because it does not need to operate MAC with zero values. Regarding BRAM consumption, the SpTFRA approach does not offer a significant advantage because of the MCSR format and the oversized memory allocation performed by EDA tools (in this case, VITIS HLS™ and Vivado™). Nevertheless, all BRAM requirements for the SpTFRA are for fully integrated memory, reducing memory access times and avoiding needing external hardware. The FFs and LUTs consumption behaves similarly for all the analyzed SpTFRA configurations; however, in all cases, these values are lower than those required by the basic TFRA.

Table 2 presents the main hardware characteristics obtained for the pre-synthesis stage of the SpTFRA implementations for the lowest resource consumption cases. The TFRA hardware implementations $HW_{1}$ and $HW_{2}$ are shown for reference only, as these are not physically implementable because they exceed 100% of the resources available on the FPGA platform used. TFRA and SpTFRA implementations execute steps i to v (see Figure 4) only once, so the latency and response time of the CeAn, We, and Co blocks become predictable.

CeAn is a common block for all hardware implementations and has a latency of 3481 clock cycles for a system clock frequency of 125 MHz. Blocks responsible for processing sparse matrices, such as coefficients calculations Co and weights vector calculation We, show differences in their latency figures due to the variation in the number of operations required for each matrix. Op is also a common block for all proposed hardware implementations and has a latency of 320,527 clock cycles per iteration. However, the total time execution of Op block depends on the number of iterations performed to evaluate the TFRA optimization functions (Equations (Equation 20) and (Equation 21)).

### 5.3. Design Verification

Figure 13 and Figure 14 represent the results of the functional validation of the proposed hardware implementations evaluated through behavioral simulation. As observed, the obtained values are close to those expected for the resultant forces and the variables μ and ϕ. These variables verified that the SpTFRA model meets the generalization and scalability criteria when applied to two tactile sensors with different resolutions, two single contact models, and different tangential force orientations.

The values presented in Table 3 were found when evaluating the system response. These include the maximum values of the estimation error, the error distribution, and the maximum response time for the TFRA and SpTFRA implementations studied. The maximum relative error in the estimation of the tangential resultant forces is 7.70%, and of the normal resultant forces is 12.57%. The maximum relative errors in the estimation of μ and ϕ were 14.67% and 1.93%, respectively. These values are close to those obtained by software implementations for the TFRA.

The obtained results verified that force reconstruction performs better in square tactile sensor arrays (Sensor 2) than rectangular ones (Sensor 1). Performed tests included normal stress input data up to 50,000 N/m^2^ to reconstruct forces with magnitudes of the resultant forces on each axis around 6 N. However, the input data can be changed without affecting the hardware implementation, allowing the reconstruction of contact forces at different scales. In this work, force magnitudes were estimated in contact areas up to $40\times 40\;{\mathrm{mm}}^{2}$. Other authors estimate triaxial forces of 1 N on contact areas of $12.5\times 12.5\;{\mathrm{mm}}^{2}$ [2] and 10 N in $14.2\times 14.2\;{\mathrm{mm}}^{2}$ [10] in applications of robotic manipulation and implemented in PC workstations. This demonstrates the capabilities of TFRA and SpTFRA to process triaxial forces with the portability, low power, and high processing power of a high-performance FPGA.

Figure 15 shows the response times obtained by simulating the behavior of the TFRA and SpTFRA implementations on an FPGA operating at a clock frequency of 125 MHz. The force estimation for SpTFRA implementations was performed in an average time of 58.68 ms, with a worst case of 78.88 ms. These results are in the same ranges as other works that report operation in real time, such as in [16] (33 ms), [23] (44 ms), and in [2] (300 ms). However, it should be noted that all those response times were obtained using software implementations running on PC workstations.

## 6. Conclusions

This work demonstrated an efficient hardware implementation for the reconstruction of triaxial contact forces in a distribution of discrete normal stresses obtained from pressure-sensor arrays based on different transduction technologies, including piezoresistive, resistive, and capacitive, as well as the potential of high-performance embedded platforms, such as FPGAs, to process contact forces in tactile sensing systems.

The presented hardware design approach leverages the generalized matrix–vector multiplication operation to optimize hardware resource consumption by replacing dense matrices with approximate sparse matrices, which is achieved using filters specifically designed for this task.

The use of high-level design tools reduces the design effort by allowing different software and hardware implementation options to be evaluated simultaneously, therefore reducing the data storage requirements, the number of processing elements, and the energy consumption of the developed system.

Future works could include exploring alternatives for data access, addressing, and reuse for TFRA and SpTFRA implementations that take advantage of the on-chip memory availability of FPGAs and other embedded systems to overcome the bottleneck caused by poor memory access times. In addition, since the total response time of the system depends on the number of iterations of the optimization algorithm (stage vi of the TFRA algorithm), it becomes very sensitive to delays in the Op block, so efforts could be made to achieve higher task concurrency and explore other minimization techniques in the optimization algorithm.

## Figures and Tables

**Figure 1 sensors-24-07829-f001:**
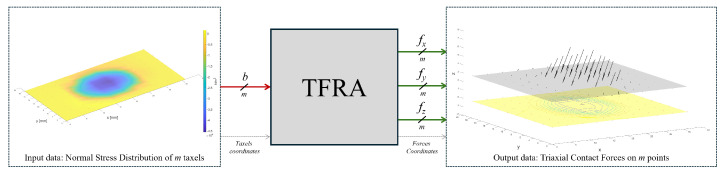
Triaxial forces reconstruction algorithm (TFRA).

**Figure 2 sensors-24-07829-f002:**
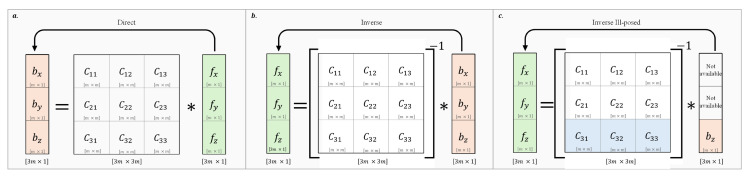
Relationship between *m*-stress values and *m*-triaxial forces for an *m*-taxel tactile sensor array under the Businessq Equation: (**a**) Direct problem, (**b**) Inverse problem, and (**c**) Ill-posed inverse problem from normal stress data.

**Figure 3 sensors-24-07829-f003:**
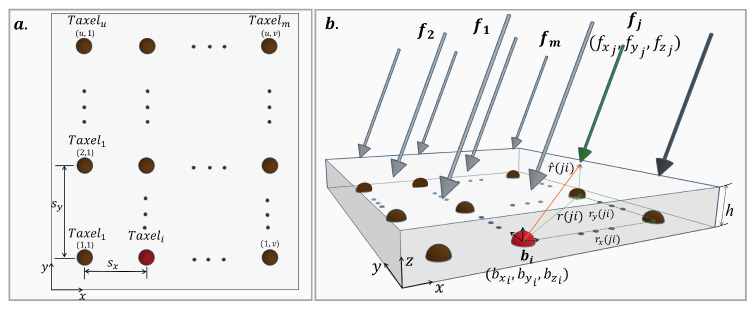
Normal stress on a tactile sensor array of *m* taxels: (**a**) Sensor top view, and (**b**) Interaction between the *i*-th taxel (dot in red) and the *j*-th force vector (green arrow) on the sensor surface for *m*-stress values and *m*-force vectors.

**Figure 4 sensors-24-07829-f004:**
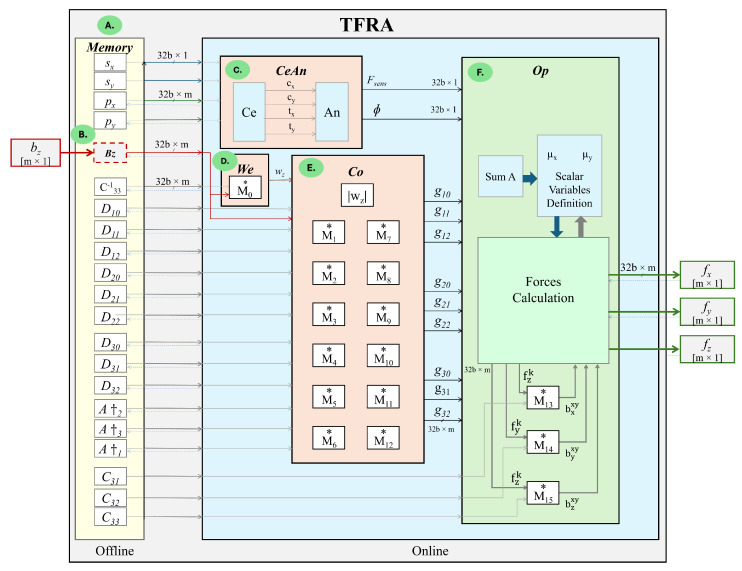
Blocks for the TFRA implementation in hardware: (**A**) ***Memory*** stores the set of matrices in memory, (**B**) ***Bz*** reads the data from the $b_{z}$ sensor in memory, (**C**) ***CeAn*** computes the contact centroids and tangential force angle, (**D**) ***We*** calculates the initial solution weight vector $w_{z}$, (**E**) ***Co*** calculates the $g_{pq}$ coefficients, and (**F**) ***Op*** finds the optimal values of the algorithm. Note that the symbol * represents a functional block in hardware that implements a matrix-vector multiplication.

**Figure 5 sensors-24-07829-f005:**
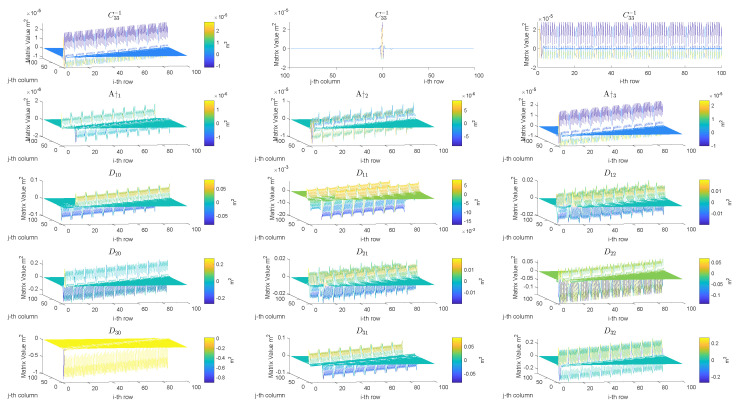
Behavior of magnitudes of the TFRA matrices (size 100×100) for a tactile sensor of 10×10 taxels. The first three graphs show the components of the matrix $C_{33}^{-1}$ for different rows or columns.

**Figure 6 sensors-24-07829-f006:**
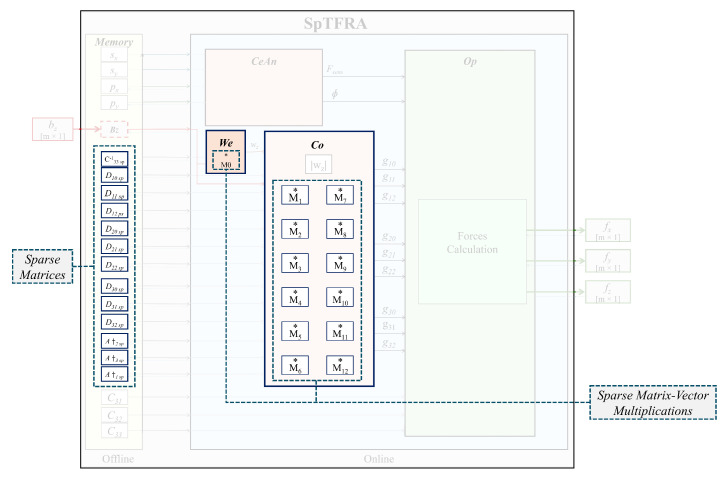
Hardware implementation of the SpTFRA algorithm. This figure highlights the blocks that are modified with respect to the original TFRA, which change from normal matrix operations to sparse matrix operations.

**Figure 7 sensors-24-07829-f007:**
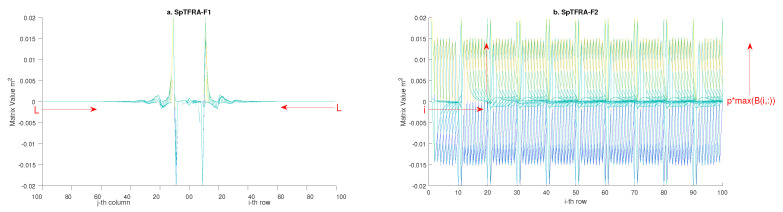
SpTFRA filters for matrix *B* applied to the: (**a**) SpTFRA−F1, selects the non-zero values closest to the diagonal; (**b**) SpTFRA−F2, selects the non-zero values greater or equal to a percentage *p* of the maximum value for each row.

**Figure 8 sensors-24-07829-f008:**
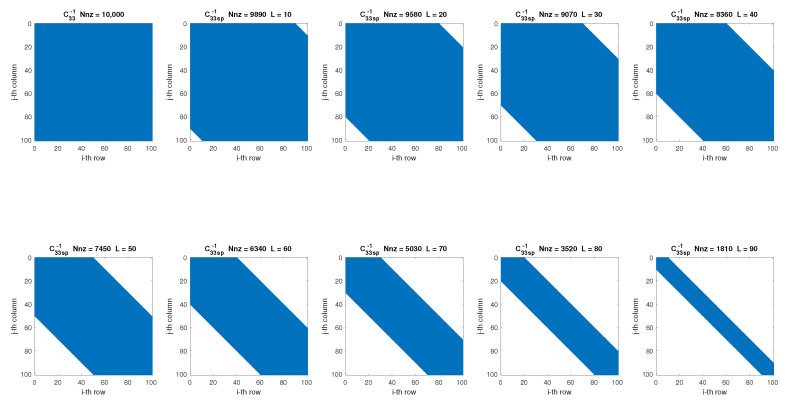
Application of the SpTFRA−F1 on the $C_{33}^{-1}$ matrix.

**Figure 9 sensors-24-07829-f009:**
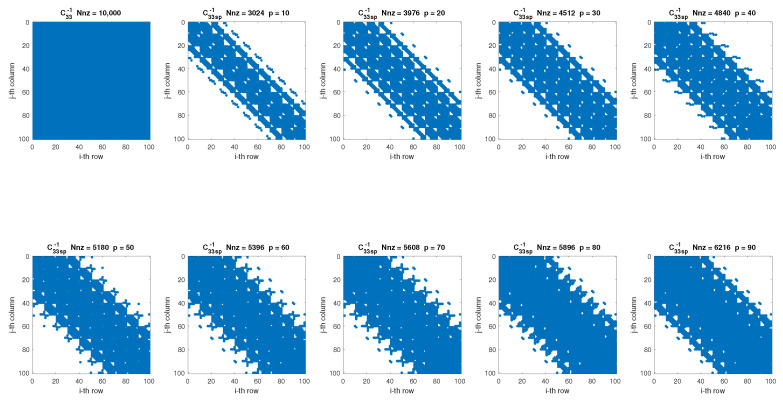
SpTFRA−F2 application on $C_{33}^{-1}$ matrix.

**Figure 10 sensors-24-07829-f010:**
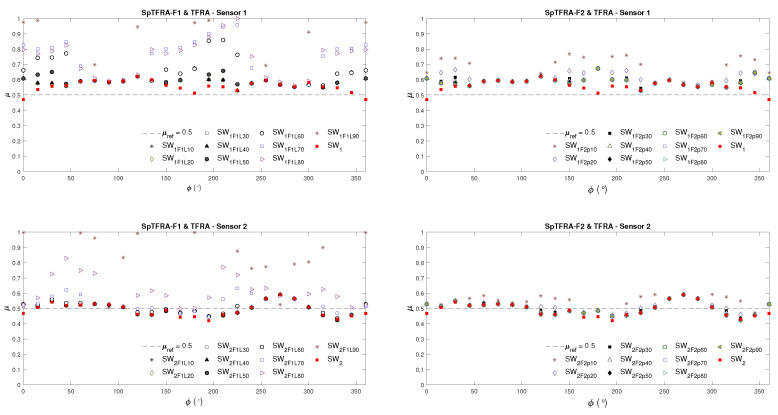
Estimated friction coefficient by applying the SpTFRA−F1 and SpTFRA−F2 filters.

**Figure 11 sensors-24-07829-f011:**
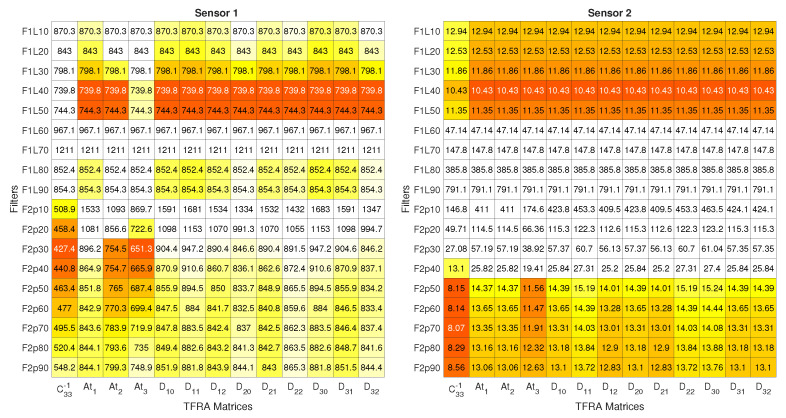
Comparative response of the filters applied to the SpTFRA model evaluating $e_{\mu }\left(B_{sp}\right)Nnz\left(B_{sp}\right)$) for each matrix. Note that the orange cells with white text represent the best cases for SpTFRA−F3.

**Figure 12 sensors-24-07829-f012:**
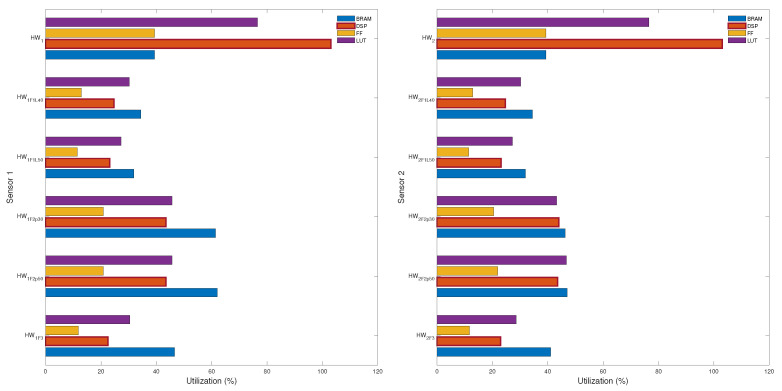
Hardware resource consumption for the TFRA ($HW_{1}$, $HW_{2}$) and SpTFRA for the two sensors analyzed.

**Figure 13 sensors-24-07829-f013:**
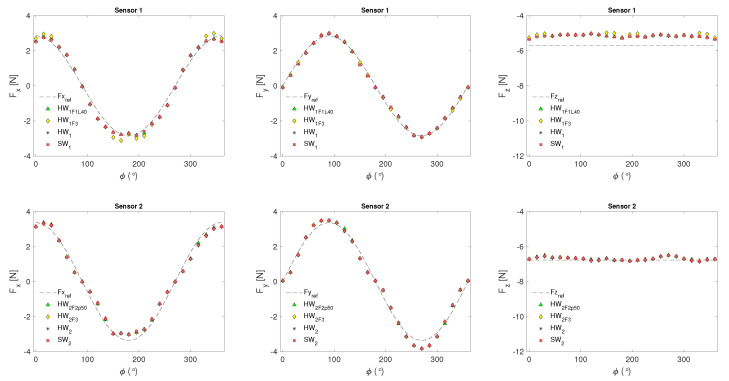
Resultant forces obtained in the TFRA and SpTFRA hardware implementations.

**Figure 14 sensors-24-07829-f014:**
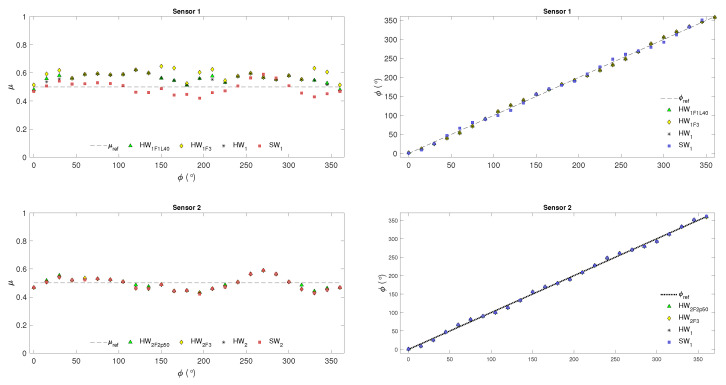
The friction coefficient and tangential force orientation results for SpTFRA hardware implementations.

**Figure 15 sensors-24-07829-f015:**
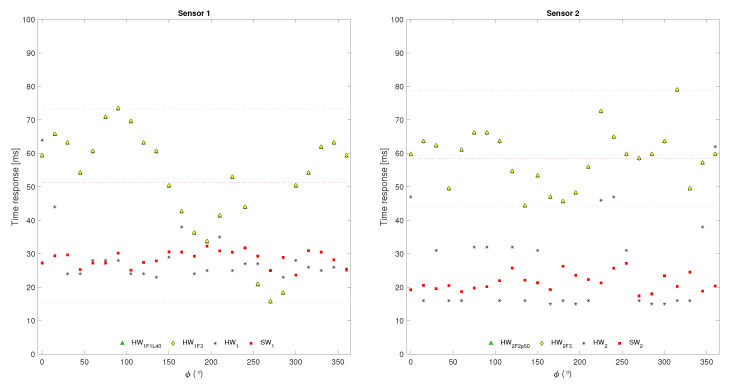
Time response for SpTFRA-HW10 and SpTFRA-HW13.

**Table 1 sensors-24-07829-t001:** Nnz values per matrix generated at the best reconstruction cases for both sensors.

Sensor	Filter	$C_{33}^{-1}$	$A_{†1}$	$A_{†2}$	$A_{†3}$	$D_{10}$	$D_{11}$	$D_{12}$	$D_{20}$	$D_{21}$	$D_{22}$	$D_{30}$	$D_{31}$	$D_{32}$	$\bar{\mu }$	HW Name
1	None	10,000	10,000	10,000	10,000	10,000	10,000	10,000	10,000	10,000	10,000	10,000	10,000	10,000	0.56	HW_1_
	F1 *L* = 40	8360	8360	8360	8360	8360	8360	8360	8360	8360	8360	8360	8360	8360	0.59	${\mathrm{HW}}_{1F1L40}$
	F1 *L* = 50	7450	7450	7450	7450	7450	7450	7450	7450	7450	7450	7450	7450	7450	0.60	${\mathrm{HW}}_{1F1L50}$
	F2 *p* = 30	4512	9462	7966	6876	9548	10,000	9400	8938	9400	9412	10,000	9550	8934	0.59	${\mathrm{HW}}_{1F2p30}$
	F2 *p* = 50	5180	9522	8552	7684	9568	10,000	9502	9320	9490	9676	10,000	9568	9326	0.59	${\mathrm{HW}}_{1F2p50}$
	F3	4512	8360	8360	6876	8360	8360	8360	8360	8360	8360	8360	8360	8360	0.49	${\mathrm{HW}}_{1F3}$
2	None	10,000	10,000	10,000	10,000	10,000	10,000	10,000	10,000	10,000	10,000	10,000	10,000	10,000	0.49	HW_2_
	F1 *L* = 40	8360	8360	8360	8360	8360	8360	8360	8360	8360	8360	8360	8360	8360	0.50	${\mathrm{HW}}_{2F1L40}$
	F1 *L* = 50	7450	7450	7450	7450	7450	7450	7450	7450	7450	7450	7450	7450	7450	0.50	${\mathrm{HW}}_{2F1L50}$
	F2 *p* = 30	4036	9294	9294	5388	9360	9932	9146	9360	9146	9932	10,000	9362	9362	0.51	${\mathrm{HW}}_{2F2p30}$
	F2 *p* = 50	5348	9430	9430	7588	9440	9968	9196	9440	9196	9968	10,000	9442	9442	0.50	${\mathrm{HW}}_{2F2p50}$
	F3	5732	8360	8360	8360	8360	8360	7395	8360	8360	8360	8360	8360	8360	0.50	${\mathrm{HW}}_{2F3}$

**Table 2 sensors-24-07829-t002:** Characteristics of the evaluated TFRA and SpTFRA hardware implementations.

Hardware	Resources Utilization				On-Chip	Latency			Throughput
	BRAM	DSP	FF	LUT	Power [W]	CeAn	We-Co	Op	[MBps]
$HW_{1}$	718	2596	215,602	209,814	3992.9 *	3481	859,480	320,527	64.67
$HW_{1F1L40}$	628	622	70,662	82,842	0.871 **		683,252		65.89
$HW_{1F3}$	849	568	64,917	83,277	0.894 **		740,387		59.96
$HW_{2}$	718	2596	215,602	209,814	3992.9 *	3481	859,480	320,527	64.67
$HW_{2F2p50}$	858	1098	120,014	128,040	0.953 **		804,205		62.72
$HW_{2F3}$	749	578	64,287	78,489	0.983 **		844,606		55.32

* Physically not implementable because this case exceeds the hardware platform capability. ** All data in this table were obtained for a junction temperature of around 26 °C.

**Table 3 sensors-24-07829-t003:** Error estimation and response time.

Hardware	Maximum Relative Error [%]						Standard Error [%]					Av. Resp. Time
	$e_{F_{x}}$	$e_{F_{y}}$	$e_{F_{z}}$	$e_{\mu }$	$e_{\phi }$		${\mathrm{SE}}_{F_{x}}$	${\mathrm{SE}}_{F_{y}}$	${\mathrm{SE}}_{F_{z}}$	${\mathrm{SE}}_{\mu }$	${\mathrm{SE}}_{\phi }$	$\bar{t_{r}}$
$SW_{1}$	5.94	3.44	10.93	9.41	1.64		2.04	1.13	1.35	3.64	0.50	28.56
$HW_{1}$	5.95	3.45	11.87	9.42	1.93		2.03	1.13	1.35	2.75	0.50	28.76
$HW_{1F1L40}$	7.45	4.79	10.83	12.15	1.93		1.93	1.07	1.27	2.77	0.50	51.27
$HW_{1F3}$	7.70	4.17	12.57	14.67	1.93		1.81	0.85	1.30	3.59	0.50	51.49
$SW_{2}$	7.84	5.44	2.69	4.46	3.16		1.73	1.84	1.02	2.82	0.69	21.53
$HW_{2}$	6.12	5.90	3.66	8.97	2.20		1.69	1.83	1.02	2.21	0.68	25.96
$HW_{2F2p50}$	5.51	5.92	3.37	9.01	2.20		1.60	1.88	1.05	2.23	0.68	58.52
$HW_{2F3}$	6.12	5.91	3.67	8.99	2.20		1.68	1.84	1.04	2.19	0.68	58.68

## Data Availability

The data used in this paper can be requested from the corresponding author upon request.

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
