# Peer review of "Hardware Implementation for Triaxial Contact-Force Estimation from Stress Tactile Sensor Arrays: An Efficient Design Approach"

_sensors, 2024, doi:10.3390/s24237829_

Round 1

Reviewer 1 Report

Comments and Suggestions for Authors

The main contribution of this paper lies in the design of an efficient tactile sensing algorithm, which utilizes the characteristics of generalized sparse matrix-vector multiplication to reduce the area, power consumption, and data storage required for real-time hardware implementation. Additionally, the paper addresses the hardware challenges of implementing a multi-axis contact force estimation algorithm on a field-programmable gate array (FPGA) development platform, employing a high-level description method for implementation. By comparing the hardware implementation results of the newly proposed sparse algorithm with existing algorithms in this work, the paper demonstrates the force vector calculation under a static single-contact model, which will aid in expanding the application of pressure sensor arrays in large-scale wearable electronic skin systems.

What are the main characteristics of the Contact Forces Reconstruction Algorithm mentioned in the paper?

How is the sparse matrix integrated into the contact force algorithm?

What efficiency criteria were considered in the hardware design process?

What type of triaxial floating-point format was used in the paper, and what is the system clock frequency?

Under what conditions are the friction coefficient test results shown in Figure 8?

What content is summarized in Figure 9?

What type of sensors does the technology or method proposed in the paper target?

What key points are mainly discussed in the conclusion section of the paper?

The authors need to emphasize the above issues before it can be considered for publication.

Reviewer 2 Report

Comments and Suggestions for Authors

The authors propose an innovative approach by utilizing sparse matrix-vector multiplication (SpTFRA) to enhance computational efficiency and scalability. However, some points need to be addressed.

  1. A clearer illustration of the SpTFRA algorithm should be provided to explain how the sparse matrices differ from traditional approaches. Additionally, a comparison with other hardware implementations or state-of-the-art tactile sensing systems should be included, highlighting the advantages and potential limitations of the SpTFRA approach.
  2. Incorporating detailed statistical evaluations, such as error distributions, would provide a more comprehensive assessment of the system’s reliability and robustness.
  3. The authors briefly mention FPGA resource constraints. However, practical challenges such as thermal management and long-term stability should be explained in detail.
  4. The implications of response time for real-time applications should be discussed. For instance, in fast-paced robotic tasks, how would the proposed system perform during high-frequency operations?
  5. Detailed metrics and comparisons for energy efficiency under different workloads would be beneficial for understanding the current approach’s suitability for energy-constrained applications.

I recommend the manuscript be revised to address these points before reconsideration for publication.

Round 2

Reviewer 2 Report

Comments and Suggestions for Authors

The revisions made by the author have significantly improved the manuscript by properly reflecting the reviewers' comments. Therefore, I agree with publishing this manuscript.